# Examining the characteristics of social and behavior change communication intervention costs in low- and middle-income countries: A hedonic method approach

Lori A. Bollinger[1], Nicole Bellows[1]*, Rachael Linder[2]

**1** Avenir Health, Glastonbury, Connecticut, United States of America, **2** Guidehouse, Atlanta, Georgia, United States of America

* nbellows@avenirhealth.org

**Data Availability Statement:** All relevant data are within the paper and its Supporting Information files.

## Abstract

Understanding the costs of health interventions is critical for generating budgets, planning and managing programs, and conducting economic evaluations to use when allocating scarce resources. Here, we utilize techniques from the hedonic pricing literature to estimate the characteristics of the costs of social and behavior change communication (SBCC) interventions, which aim to improve health-seeking behaviors and important intermediate determinants to behavior change. SBCC encompasses a wide range of interventions including mass media (e.g., radio, television), mid media (e.g., community announcements, live dramas), digital media (e.g., short message service/phone reminders, social media), interpersonal communication (e.g., individual or group counseling), and provider-based SBCC interventions focused on improving provider attitudes and provider-client communication. While studies have reported on the costs of specific SBCC interventions in low- and middle-income countries, little has been done to examine SBCC costs across multiple studies and interventions. We use compiled data across multiple SBCC intervention types, health areas, and low- and middle-income countries to explore the characteristics of the costs of SBCC interventions. Despite the wide variation seen in the unit cost data, we can explain between 63 and 97 percent of total variance and identify a statistically significant set of characteristics (e.g., health area) for media and interpersonal communication interventions. Intervention intensity is an important determinant for both media and interpersonal communication, with costs increasing as intervention intensity increases; other important characteristics for media interventions include intervention subtype, target population group, and country income as measured by per capita Gross National Income. Important characteristics for interpersonal communication interventions include health area, intervention subtype, target population group and geographic scope.

**Funding:** Breakthrough RESEARCH is made possible by the generous support of the American people through the United States Agency for International Development (USAID) under the terms of cooperative agreement no. AID-OAA-A-17-00018. The contents of this document are the sole responsibility of the Breakthrough RESEARCH and Population Council and do not necessarily reflect the views of USAID or the United States Government. The funders had no role in study design, data collection and analysis, decision to publish, or preparation of the manuscript.

**Competing interests:** The authors have declared that no competing interests exist.

## Introduction

Understanding the costs of health interventions is critical for generating budgets, planning programs, and conducting economic evaluations to use when allocating scarce resources. Costing studies in global health usually examine underlying cost drivers for health-related interventions using disaggregated data (e.g., personnel costs) [1, 2], although some also include various characteristics (e.g., geographic scope) as seen in hedonic pricing models [3, 4]. Hedonic pricing models assume that a price and the characteristics of the good or service under analysis define a set of implicit or "hedonic" prices, where the price is a function of the good/service characteristics [5, 6]. The approach is used in a wide variety of applications, including to value characteristics of housing in China [7]; urban green space in Stockholm [8]; and even animal shelters in Michigan [9]. The United States Bureau of Labor Statistics routinely uses hedonic pricing models to estimate inflation rates for over 270 goods [10]. In the health sector, hedonic pricing models have been used to price new drugs in Germany [11] and examine drivers of hospital costs [12].

Here, because disaggregated costing data are not readily available, we utilize techniques from the hedonic pricing literature to estimate the characteristics of the costs of social and behavior change communication (SBCC) interventions, a somewhat neglected area for costing in global health. SBCC interventions aim to improve health-seeking behaviors and important intermediate determinants to behavior change, including knowledge, attitudes, communication, and social norms [13]. SBCC interventions encompass a wide range of interventions including mass media (e.g., radio, television (TV)), mid media (e.g., community announcements, live dramas), digital media (e.g., short message service (SMS)/phone reminders, social media), interpersonal communication (IPC) (e.g., individual or group counseling), and provider-based SBCC interventions focused on improving provider attitudes and provider-client communication [14]. The success of these interventions is critical in responding to health needs and ensuring the overall success of global health interventions [15, 16].

While there are some studies reporting the costs of SBCC interventions in low- and middle-income countries, little has been done to examine SBCC costs across multiple studies, interventions, and health areas. A recent review compiled and examined the existing literature on SBCC intervention costs [17] and made the data accessible to others via the SBCC Cost Repository [18] and the Global Health Cost Consortium's Unit Cost Study Repository (UCSR) [19].

Building on this work, this analysis utilizes SBCC unit costs to explore the relationship between a set of underlying characteristics of SBCC interventions and their costs, as reported in the literature. This understanding would enable one to estimate more specific unit SBCC cost estimates based on the intervention's characteristics, thus improving SBCC budgeting and program planning.

## Materials and methods

### Materials

The peer-reviewed SBCC cost literature was searched using PubMed in May 2018 and again in September 2019, with further studies added from the grey literature and secondary sources. We included 147 cost studies with a total of 847 cost observations, of which 355 cost observations were unit costs and 120 of these unit costs were comparable enough for synthesis purposes. SBCC intervention categories included were interpersonal communication, both individual and group (31%); mass media (21%); packages delivering more than one SBCC intervention category but reported as a combined cost observation (15%); other SBCC

intervention types (13%); and studies with more than one SBC intervention category with each reported as a distinct cost observation (20%). Median unit costs varied by SBC intervention, driven largely by differences in the unit of analysis and other contextual factors. We updated the dataset in 2020 and 2021. For further information on the literature search, see [17], while the full dataset is available at [18].

From this dataset, we utilize SBCC unit cost estimates that are comparable enough for analytical purposes, i.e., reporting consistent units of measurement for either (1) per person exposed; (2) per person participating; or (3) per provider for provider-based interventions. In addition, these estimates all had a provider costing perspective [20]; measured costs during either the implementation phase or overall total costs; and have reasonable data quality as measured by internal consistencies within the reported data. The estimates contain data for several types of SBCC interventions, including media; IPC; and provider-based interventions; and for multiple health areas, including family planning/reproductive health (FP/RH); HIV; malaria; and maternal, newborn and child health (MNCH).

The potential list of SBCC intervention characteristics hypothesized to be statistically related to the reported unit costs of the intervention includes a series of categorical variables for the health areas and SBCC interventions listed above, with media further disaggregated into print, mid media, SMS, TV, radio, and mixed media subtypes, and IPC further disaggregated into individual, group, and IPC of either type implemented in conjunction with other SBCC activities, which include other SBCC interventions (e.g., print). The potential effects of technological change are explored using year of study publication (a continuous variable) and also three categorical variables related to study publication date: <2005, 2005–2012, and >2012 (see Table 1). The cutoffs are based on changes in the types of interventions reported on, where group IPC studies became prevalent beginning in 2005 and SMS studies became prevalent beginning in 2013. Potential socioeconomic effects on the costs of interventions are explored using gross national income per capita (GNIpc), also a continuous variable, and region where the intervention was implemented. Further categorical variables are defined for key characteristics [20] including intervention ownership, geography, urbanicity, population targeted by the intervention, region, type of cost (economic vs. financial), as well as whether significant data quality issues were noted in the reporting of the data. Data quality issues included concerns such as small sample sizes, lack of clarity in defining the denominator of the unit cost, and potentially missing cost information.

**Table 1. List of SBCC intervention characteristics and associated variables.**

| Characteristic | Associated Variables |
| --- | --- |
| Health areas | FP/RH, HIV, malaria, MNCH, other |
| Intervention subtypes | Media: print, mid media, SMS, TV, radio, and mixed<br>IPC: individual, group, and mixed IPC |
| Technological change | Year of study publication<br>Date of study publication <2005, 2005–2012, >2012 |
| Socioeconomic status | Gross national income per capita (GNIpc)<br>Region: Sub-Saharan Africa, Asia, Latin America, other |
| Intervention Ownership | Public, private, non-governmental organization (NGO), other |
| Geographic scope | National, district, local, other |
| Urbanicity | Urban, rural, mixed |
| Population | General, youth, at-risk |
| Type of cost | Economic, financial |
| Data quality issues | Yes, no |
| Intervention intensity | High, medium, low |

In addition, we created an "Intervention intensity" variable, which was then assigned to each observation. The intensity designations were made within the intervention subtypes and did not account for different intensity across subtypes (e.g., mass media vs. IPC). For mass media and mid-media interventions, a low-intensity intervention had one to three brief messages that used one mode of distribution (e.g., radio or print) and was conducted for a limited time period (e.g., 3 months or less). In contrast, a high-intensity intervention involved a longer and more complex development process, such as a serial television program with multiple episodes and characters that ran for two years. Medium-intensity interventions were more complex than low-intensity ones but did not rise to the criteria of high-intensity, such as a campaign that developed several different spots with accompanying billboards that ran for six months. For mobile digital interventions, those that were interactive, had multiple touchpoints with a client, and had a considerable development process were deemed high-intensity, while those with a simple SMS appointment reminder were deemed low-intensity. For IPC, low-intensity interventions were those with only one interaction and implemented by non-professionals. In group IPC contexts, a low-intensity intervention generally had a large number of participants within a single group (e.g., >20) whereas high-intensity IPC interventions used trained professionals and had numerous contacts with the same person. The extent to which IPC interventions developed curricula for training counselors was also a factor in determining intensity. Two researchers independently classified each intervention applying the above criteria to the SBCC intervention description in the study; when the classifications differed, the study was discussed, and a senior researcher assigned the final classification.

## Methods

We used Stata 16.0 to estimate a series of Ordinary Least Squares (OLS) models to examine the relationship between unit costs of SBCC interventions and various characteristics of the interventions. We tested both levels and natural logarithms of unit costs as dependent variables; because the initial OLS results sometimes predicted negative costs, we also tested a Tobit model, where the dependent variable is a percentage of the median cost for a particular intervention subtype (e.g., an SBCC print intervention unit cost in India is x% of the SBCC print intervention median unit cost). We also tested both levels and natural logarithms of the GNIpc independent variable, the only continuous independent variable in the regressions. We tested and corrected for heteroskedasticity as necessary, using robust standard errors. We also tested for multicollinearity using the variance inflation factor (VIF); because our ultimate objective was extrapolation, however, we were able to utilize the (unbiased and consistent) coefficients even if multicollinearity was present [21].

Because of sample size issues, we prioritized the inclusion of several independent variables: health areas, intervention subtypes, and socioeconomic status. We explored the inclusion of different independent variables (from Table 1) using t-tests for individual independent variables and F-tests for groups of categorical variables, particularly given the multicollinearity issues [22]. We evaluated the different specifications based on these criteria as well as other criteria, including comparing the R-squared values (calculated correctly for the Tobit specification [23]), which calculates how much of the variation in SBCC unit costs is explained by the SBCC intervention characteristics, and F-tests for the overall regressions, which tests the joint significance of the set of independent variables.

The final model specification tested is:

$$UC_{ij} = \alpha + (\beta_1 \times HA_{ij}) + (\beta_2 \times IS_{ij}) + (\beta_3 \times T_i) + (\beta_4 \times SES_j) + (\beta_{5k} \times C_{ijk}) + (\beta_6 \times DQ_{ij})$$

where

- $UC_{ij}$ is the average cost per outcome for intervention $i$ located in country $j$. (either levels, natural logarithms, or percent of the intervention subtype median unit cost);

- $HA_{ij}$ is a series of categorical variables for Health Areas, which take the value 1 when the Health Area that intervention i in country j focuses on is either FP/RH, HIV, Malaria, MNCH, or Other, and is zero otherwise (reference category: HIV);

- $IS_{ij}$ is a series of categorical variables for Intervention Subtypes, which take the value 1 when the Intervention Subtype for intervention i in country j focuses on either print, mid media, SMS, TV/radio, or mixed for the Media regression, and either individual, group, or mixed IPC for the IPC regression, and is zero otherwise (reference category: TV/radio);

- $T_i$ is a variable measuring Technology, and is represented in one of two ways: either Time, defined as the year of the publication of intervention i results, or a series of categorical variables which take the value 1 when the date of publication is either before 2005, between 2005 and 2012, or after 2012, and is zero otherwise (reference category: before 2005);

- $SES_j$ is a variable measuring Socioeconomic Status, and is represented in one of two ways: either GNIpc (levels or natural logarithms) for country j, or a series of categorical variables which take the value 1 when the region of country j is either Sub-Saharan Africa, Asia, Latin America or Other, and is zero otherwise (reference category: Sub-Saharan Africa);

- $C_{ijk}$ is a series of categorical variables for Characteristics measuring various k characteristics of intervention i in country j, where the categorical variables within each k characteristic take the value 1 when the intervention has that characteristic, and zero otherwise. The k sets of characteristics (list of categories; reference category) are intervention ownership (public, NGO, private, other: reference category: public), geographic scope (local, district, national, other; reference category: local), urbanicity (urban, rural, mixed; reference category: urban), population (general/youth, at-risk; reference category: general/youth), type of cost (economic or financial; reference category: economic), and intensity of intervention (low, medium, high; reference category: low);

- $DQ_{ij}$ equals one if there are significant Data Quality issues for the observation, and zero if there are no significant data quality issues.

## Results

In total, 157 observations were included in the initial analysis: 66 observations for media interventions, 79 for IPC interventions, and 12 for provider SBCC interventions. Table 2 displays the median unit costs for each intervention subtype as well as their standard deviations and ranges:

As expected, the median unit costs are lowest for the media SBCC interventions, which range from $0.12 for television and print to $0.93 for SMS/phone-based interventions per person exposed. In contrast, the median unit costs for IPC interventions are more similar, ranging from $6.26 per person participating in group IPC to $6.57 per person participating in IPC provided in conjunction with other SBCC interventions. Provider SBCC has the highest median unit cost of $125.73 per provider trained. Within each intervention, the range of reported unit costs varies substantially, with media interventions ranging from $0.01 to over $4.00 per person exposed and the IPC unit costs ranging from $0.15 to $67.75 for group IPC per person participating. Unit costs for provider training have the widest range, varying from a low of $1.23 per provider trained to $2,603.49 per provider trained. The wide dispersion of costs can be

**Table 2. Median and range of unit costs by intervention subtype.**

| Intervention | Observations | Median unit cost | Standard deviation | Unit cost range |
|---|---|---|---|---|
| *Media–per person exposed* | | | | |
| Radio | 11 | $0.28 | $1.53 | $0.01 - $4.36 |
| Television | 11 | $0.12 | $0.24 | $0.01 - $0.78 |
| Print | 7 | $0.12 | $0.38 | $0.02 - $1.08 |
| Mixed mass media | 10 | $0.38 | $0.59 | $0.01 - $2.01 |
| Mid media | 11 | $0.41 | $1.41 | $0.15 - $3.75 |
| SMS/phone | 16 | $0.93 | $18.61 | $0.03 - $4.42 |
| *Interpersonal communication–per person participating* | | | | |
| Individual IPC | 18 | $6.40 | $23.31 | $0.35 - $79.23 |
| Group IPC | 34 | $6.26 | $12.64 | $0.15 - $67.75 |
| IPC + other SBCC interventions | 27 | $6.57 | $11.40 | $0.78 - $40.27 |
| *Provider-based SBCC–per provider trained* | | | | |
| Provider communication training | 12 | $125.73 | $728.39 | $1.23 - $2,603.49 |

seen even more clearly when displayed using charts (Figs 1–3); upon examination, there does not seem to be any systematic or non-random pattern.

It quickly became clear during the estimation process that the effect of the underlying characteristics of costs varied substantially by main SBCC intervention type; thus, we proceeded with estimating separate equations for media-focused SBCC interventions and for IPC-focused SBCC interventions, dropping provider-based SBCC interventions due to its small sample size

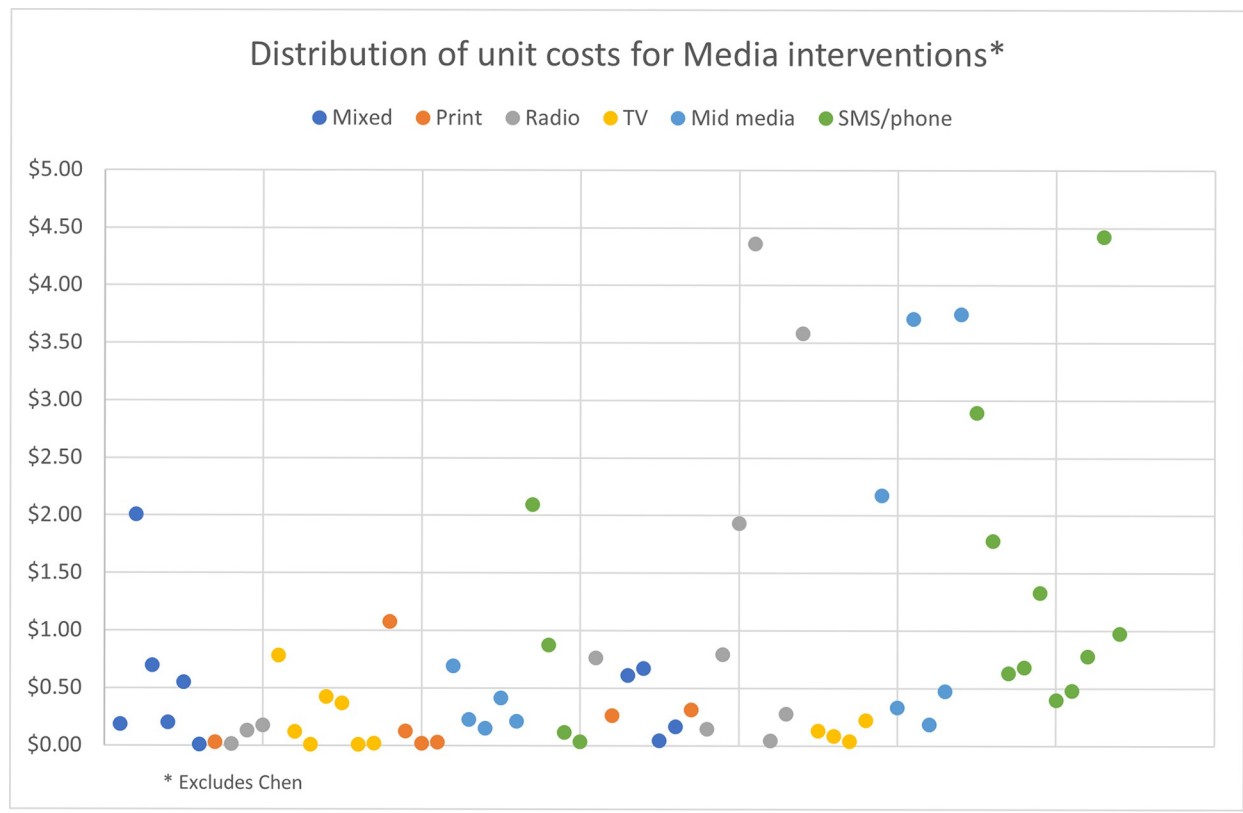

**Fig 1. Distribution of unit costs for media interventions.**

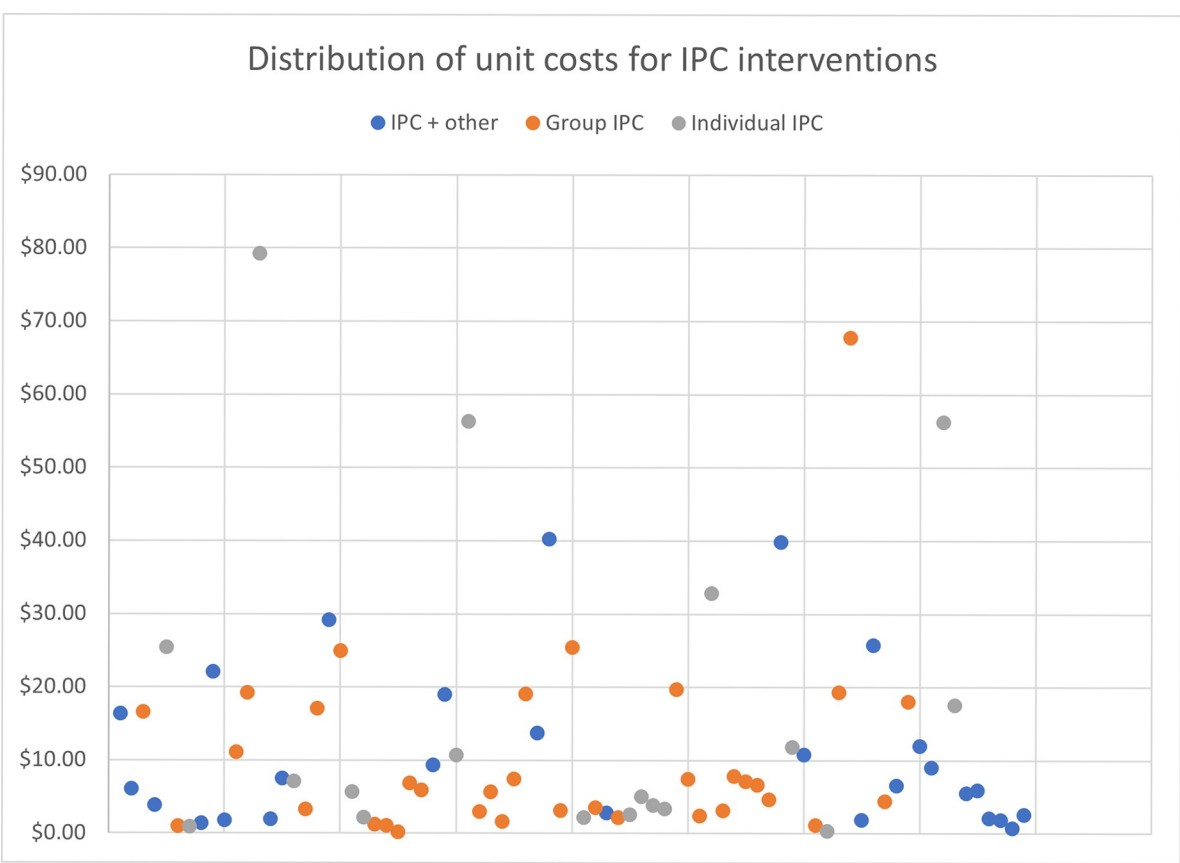

**Fig 2. Distribution of unit costs for Interpersonal Communication (IPC) interventions.**

(n = 12). In addition, several of the initial independent variables specified were consistently of very low statistical significance and thus were dropped from the analysis: the variables related to study date; the categorical variable measuring data quality; and whether the unit cost was reported in economic or financial terms. Descriptive statistics by main intervention type for the final regression variables tested in the final models are presented in Table 3.

In both samples, the majority of the observations are found in the FP/RH, HIV, and MNCH health areas; for media interventions the highest percentage for health areas was in FP/RH (30.3%) while for IPC interventions the highest percentage was in the MNCH health area (31.6%). Note that very few malaria SBCC costing studies are included in this analysis, as the studies typically report the cost per insecticide treated net distributed vs. per person exposed/participated, the outcome used here (5% for both media and IPC samples). Geographically, most observations are from Sub-Saharan Africa and Asia, with a stronger Sub-Saharan Africa representation for the IPC interventions than media interventions (57% vs. 48.5%). One key difference between the two datasets is the intervention scale, where more media interventions were implemented nationally than IPC interventions (39.4% vs. 3.8%). Finally, more media interventions were of low intensity than IPC interventions (37.9% vs. 25.3%).

Regression results for the final specifications for both the Ordinary Least Squares (OLS) and the Tobit models, each displayed separately for Media and IPC unit costs, are shown in Table 4. The final dependent variable for the OLS results for Media interventions is the natural logarithm of unit cost; for the OLS results for IPC interventions, it is the linear unit cost; and for both Tobit sets of results, it is the linear percent of the intervention subtype median unit

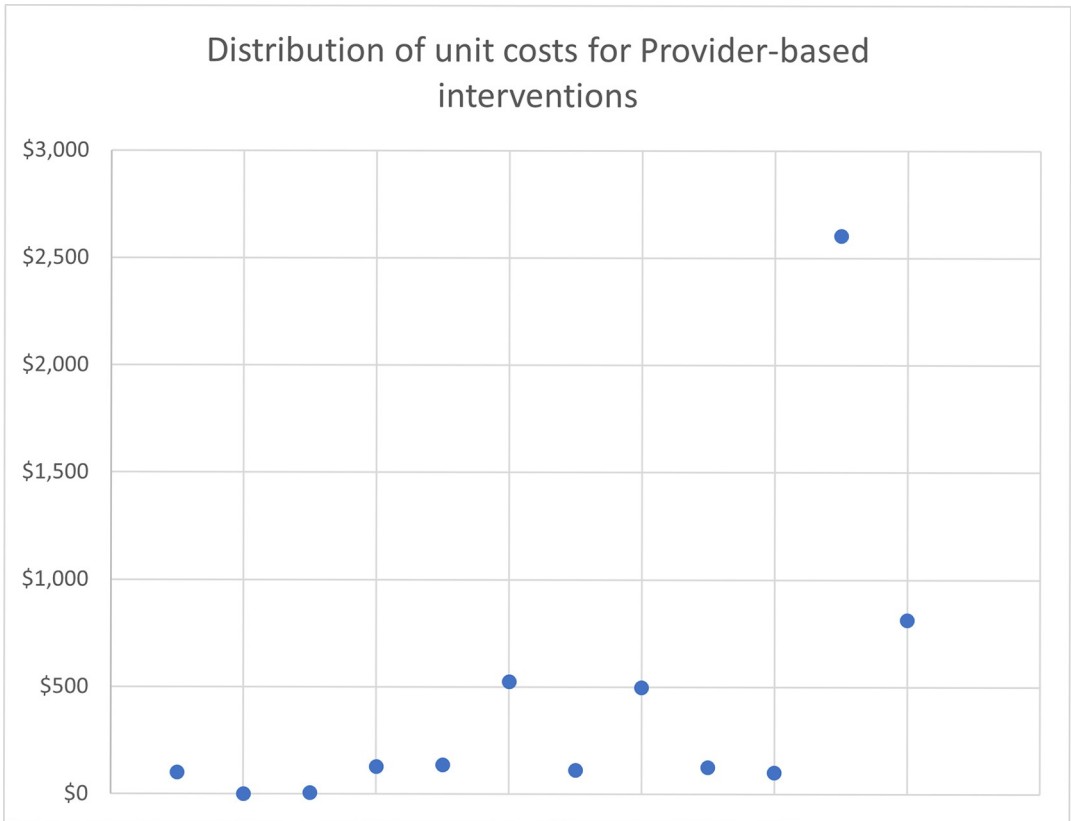

**Fig 3. Distribution of unit costs for provider-based interventions.**

cost. As noted above, moderate multicollinearity [24] is present in the dataset; the VIFs range from 1.15 to 2.76 with a mean VIF of 1.85 (see S1 Table for detailed regression results including regression-specific VIFs). This multicollinearity sometimes required that the statistical significance of categorical variables be tested as a group; if the statistical significance of one or more of the individual categorical variables within a group was below 10%, then the statistical significance of the entire group was jointly tested using an F-test. These results are listed as "*Group*" in relevant rows in the regression table, where the number of asterisks indicate the statistical significance of the F-test for the entire group of categorical variables. Note that all independent variables are categorical variables with the exception of GNIpc.

The OLS model explains over 71 percent of total variance in the unit costs of media. Each intervention intensity variable is significant at the $p < 0.01$ level, with medium and higher intensity interventions costing more than lower intensity interventions, along with two of the four intervention subtypes, mid media and SMS/phone; both costing more than the reference category, TV/radio, and the variable indicating an at-risk target population group, also costing more than the reference category, a combined general/youth target population. Note that the group of intervention subtype variables are jointly significant at the $p < 0.01$ level as well. The natural logarithm of the GNIpc variable is statistically significant at $p < 0.05$, while two of the ownership variables are statistically significant at $p < 0.1$, NGO and Other, with all three jointly significant at $p < 0.1$, all showing higher prices relative to the reference category, public ownership. Two of the three geographic scope variables, District and Other, are significant at $p < 0.1$, while the group of geographic scope variables is jointly significant at $p < 0.1$, all costing less than geographic scopes that are local.

**Table 3. Descriptive statistics of regression variables, by main intervention type.**

| | Media | IPC |
|---|---|---|
| Observations | 66 | 79 |
| **Health Areas** | | |
| FP/RH (%) | 30.3 | 19.0 |
| HIV (%) | 28.8 | 26.6 |
| Malaria (%) | 4.5 | 5.1 |
| MNCH (%) | 19.7 | 31.6 |
| Other (%) | 16.7 | 17.7 |
| **Intervention sub-types** | | |
| Print (%) | 3.0 | |
| Mid media (%) | 24.2 | |
| SMS/phone (%) | 24.2 | |
| TV/Radio (%) | 33.3 | |
| Mixed (%) | 15.2 | |
| Individual IPC (%) | | 22.8 |
| Group IPC (%) | | 43.0 |
| Mixed IPC (%) | | 34.2 |
| **Socioeconomic status** | | |
| Average GNIpc (USD) | 2,333 | 2,763 |
| Sub-Saharan Africa (%) | 48.5 | 57.0 |
| Asia (%) | 45.5 | 27.8 |
| Latin America (%) | 3.0 | 11.4 |
| Other (%) | 3.0 | 3.8 |
| **Ownership** | | |
| Public (%) | 47.0 | 38.0 |
| Private (%) | 7.6 | 10.1 |
| NGO (%) | 19.7 | 22.8 |
| Other (%) | 25.8 | 29.1 |
| **Geographic Scope** | | |
| National (%) | 39.4 | 3.8 |
| District (%) | 40.9 | 25.3 |
| Local (%) | 10.6 | 58.2 |
| Other (%) | 9.1 | 12.7 |
| **Urbanicity** | | |
| Urban (%) | 27.3 | 25.3 |
| Rural (%) | 18.2 | 25.3 |
| Mixed (%) | 54.5 | 49.4 |
| **Population** | | |
| General/Youth (%) | 86.4 | 88.6 |
| At-risk (%) | 13.6 | 11.4 |
| **Intensity of intervention** | | |
| High (%) | 18.2 | 24.1 |
| Medium (%) | 43.9 | 50.6 |
| Low (%) | 37.9 | 25.3 |
| **Dependent variables** | | |
| Unit cost (USD 2020) | 2.42 | 12.31 |
| Ln(Unit cost) | -1.08 | 1.84 |
| Percent of median unit cost | 3.78 | 1.86 |

**Table 4. Regression results on the characteristics of media and IPC intervention unit costs.**

| | Media | | | | Interpersonal communication (IPC) | | | |
|---|---|---|---|---|---|---|---|---|
| Variables | 1. OLS | | 2. Tobit | | 3. OLS | | 4. Tobit | |
| | Coeff. | Std. Error | Coeff. | Std. Error | Coeff. | Std. Error | Coeff. | Std. Error |
| Intensity of intervention (ref: Low) | | | | | | | | |
| High | 2.69*** | 0.39 | 5.27*** | 1.11 | 29.2*** | 4.70 | 4.14*** | 0.70 |
| Medium | 1.58*** | 0.30 | 1.57** | 0.79 | 12.59*** | 3.16 | 1.56*** | 0.38 |
| Health Areas (ref: HIV) | | | Group* | | | | | |
| FP/RH | | | -2.48*** | 0.90 | -15.15*** | 5.66 | -2.05** | 0.83 |
| MNCH | | | 1.08 | 0.99 | -6.16** | 2.80 | -0.87** | 0.38 |
| Malaria | | | -1.83 | 1.53 | -24.39*** | 7.14 | -3.7*** | 0.95 |
| Other | | | 0.89 | 1.52 | -12.75*** | 4.20 | -1.62*** | 0.59 |
| Intervention subtypes (ref: TV/Radio, Group IPC) | Group*** | | Group*** | | | | | |
| Print | 0.23 | 0.44 | -0.77 | 0.94 | | | | |
| Mid media | 1.54*** | 0.51 | 0.82 | 0.92 | | | | |
| SMS/phone | 1.77*** | 0.48 | -0.44 | 0.92 | | | | |
| Mixed | -0.62 | 0.59 | -3.35*** | 1.01 | Group** | | Group** | |
| Individual IPC | | | | | 8.36** | 3.90 | 1.11** | 0.56 |
| Mixed IPC | | | | | -1.49 | 3.34 | -0.18 | 0.46 |
| Population (ref: General/youth) | | | | | | | | |
| At-risk | 1.19*** | 0.42 | | | 10.11*** | 5.40 | 1.84*** | 0.70 |
| Socioeconomic status | | | | | | | | |
| GNIpc | | | | | | | 0.20 | 0.17 |
| Ln(GNIpc) | -0.52** | 0.24 | 0.89* | 0.51 | 3.88** | 1.60 | | |
| Ownership (ref: Public) | Group* | | Group*** | | Group** | | Group** | |
| NGO | 0.7* | 0.40 | 2.28** | 1.03 | 23.87*** | 6.76 | 3.2*** | 0.95 |
| Private | 1.56 | 1.03 | 2.53*** | 0.77 | 6.83 | 4.73 | 0.78 | 0.70 |
| Other | 0.71* | 0.35 | 0.96 | 0.72 | 7.13** | 3.57 | 0.88* | 0.50 |
| Geographic Scope (ref: Local) | Group* | | Group* | | | | | |
| District | -0.81* | 0.45 | -1.07 | 1.25 | 8.73** | 3.72 | 1.05** | 0.51 |
| National | -0.03 | 0.50 | 0.88 | 1.32 | -17.69*** | 5.87 | -2.78*** | 0.73 |
| Other | -1.01* | 0.59 | -1.89 | 1.48 | -13.26*** | 5.77 | -1.99*** | 0.77 |
| Constant | 0.81 | | 1.55 | | -35.47 | | -0.68 | |
| Observations (Number of unit costs) | 66 | | 66 | | 79 | | 79 | |
| R-Squared[3] | 0.713 | | 0.965 | | 0.629 | | 0.772 | |
| F-statistic | 32.74*** | | 7.18*** | | 5.43*** | | 7.07*** | |

OLS, ordinary least squares regression; FP/RH, family planning/reproductive health; MNCH, maternal, neonatal, and child health; IPC, interpersonal communication; GNIpc gross national income per capita; NGO, nongovernmental organization.

Significant levels:

***$p<0.01$

**$p<0.05$

*$p<0.1$.

The dependent variable for media/OLS is the natural logarithm of unit cost; for IPC/OLS the depending variable is the linear cost; and percent of intervention subtype median unit costs for both Tobit models. "Group" indicates an F-test for the entire group of categorical variables.

The Tobit model results for media unit costs explains 97 percent of total variation, and there are some similar patterns in the explanatory variables as for the OLS model: intervention intensity remains highly statistically significant, with both medium- and high-intensity linearly

increasing relative to low-intensity interventions. In addition, Public is still the lowest cost for type of ownership, and Local scope still has the highest cost for geographic scope. Other results, however, are different: the set of health area categorical variables are now jointly significant, with a mix of relationships to the reference category, HIV. Although the group of intervention subtype variables is jointly significant at $p<0.01$, the relationship to the reference category, TV/radio, is different, with relatively lower costs now seen for Print, SMS/phone, and Mixed subtypes. Finally, the coefficient for the at-risk target population group is no longer statistically significant.

Turning to the regressions for IPC interventions, the overall performance of both models is strong, explaining between 63 and 77 percent of total variance using the OLS and Tobit models, respectively, with many highly statistically significant independent variables in each model. Here, the results for the independent variables are consistent across the two models: low-intensity interventions are less expensive relative to medium- and high-intensity interventions, and both coefficients are statistically significant at $p<0.01$. The coefficients for health areas are individually statistically significant, and all show that the highest cost for IPC interventions is observed in HIV, followed by Malaria, FP/RH, Other and then the MNCH health area. Individual IPC interventions are more expensive than group interventions, with Mixed IPC the least expensive intervention subtype. Public ownership is the least expensive type of ownership for IPC interventions, while it costs more to reach the Local and District levels, as well as at-risk population groups.

## Discussion

After compiling the existing SBCC costing literature, we utilized a hedonic method approach to estimate how SBCC intervention unit costs are related to their underlying characteristics. There are three key findings from this analysis. First, while the datasets are relatively small with 66 and 79 observations each for media and IPC interventions, respectively, and the data are widely dispersed, underlying intervention characteristics explain between 71 and 97 percent of the total variation in the unit costs of media interventions, and between 63 and 77 percent of the total variation in the unit costs of IPC interventions, and the F-tests for including all independent variables are all statistically significant at the one percent level for all models. As such, it is clear that the variability in SBCC unit costs can be explained through an analysis of these underlying SBCC intervention characteristics (e.g., intervention intensity, subtype, health area, location, ownership, and target population).

Second, among the variables explored in this analysis, intervention intensity was the clear primary driver for both media and IPC interventions, given the high statistical significance across all models. As described in the methods, the intensity variable considered a variety of factors, including the amount of effort that went into the development of the interventions and number of times an individual was exposed to or participated in the intervention. Compared to the low-intensity category, both the average- and high-intensity variables were highly statistically significant and were both associated with higher SBCC unit costs. This result is obtained despite the difficulty in generating a comprehensive rubric that captures all the dimensions of "intensity" that could be easily applied to every study, due to the heterogeneity of SBCC interventions and the different levels of reporting detail offered in SBCC costing studies. As such, we recommend that a more in-depth analysis of intervention intensity utilizing techniques, such as factor analysis or principal components analysis, be implemented to construct a validated metric for future use. At present, the findings show that a more intense SBCC intervention will be more expensive and thus program planners should budget accordingly, allowing for more funding for more intensive approaches.

A third key finding is that, outside of intervention intensity, there are important differences in which SBCC intervention characteristics are associated with unit costs based on whether the intervention is for media or IPC interventions. This finding makes sense in that these two types of SBCC interventions are quite different in how they are implemented, despite having the same goal of improving health behaviors. For example, because IPC interventions are more labor intensive, and thus labor costs make up a larger proportion of total costs, it makes sense that the GNIpc variable would be positively associated with SBCC unit costs when comparing low-income countries to lower-middle or upper-middle income countries. In contrast, SBCC media interventions would not necessarily be more expensive in upper-middle income countries because labor costs are not as large of a component of overall media SBCC costs [18]. Additionally, individuals living in upper-middle income countries generally have greater access to media via radio, television, and mobile phones, so one might expect that the unit costs of SBCC media interventions implemented utilizing these platforms would be lower since interventions would be able to reach individuals more easily.

There are notable limitations to this analysis, starting with the fact that relatively small datasets were used in this analysis. While more SBCC costing studies exist, they were not utilized in this analysis because they did not report results using a comparable denominator, but instead reported unit costs such as SBCC cost per person utilizing desired health goods/services or health outcomes [25, 26]. While these types of unit costs are also interesting and useful, going forward the field should encourage including consistently reported denominators for all SBCC costing studies (e.g., per person exposed, per person participating) in order to maximize the ability to utilize these data in analyses. The robustness of this analysis would certainly benefit from a larger sample; there is a potential for over-fitting due to the small sample size and many predictors, perhaps reflected in the high R-squared values.

Another limitation of the data is that many SBCC costing studies do not provide enough details about the intervention, leaving gaps in the data where characteristics are classified as "unspecified", particularly for intervention ownership, urbanicity, and scale. In addition, describing the development of the SBCC intervention as part of publications describing the results of the intervention would also be useful for assigning the intervention intensity; often only one dimension of intensity was available for classification purposes, which could have resulted in incorrect classifications. Similarly, because costing studies utilize different costing methodologies and include different components, a more complete description of these details in publications would facilitate future analyses. We recommend that future SBCC costing studies follow recommended published guidelines for both conducting SBCC costing studies and reporting the results [27].

Despite these limitations, it is encouraging that we can glean useful lessons on the ways in which SBCC intervention costs vary using the available SBCC unit cost data, and how differences in SBCC intervention characteristics and the geographic scope in which they are implemented influence the unit costs. The results from this analysis can be used to help calibrate budget expectations for funders and program implementers. Leveraging the analysis here, we have built an SBCC costing tool, which can be used to predict the costs of SBCC interventions for any developing country and set of characteristics included in these regression results (e.g., intervention type, intensity) [28], for use in planning and budgeting. Note that, while estimating costs is important for planning and budgeting processes, the most appropriate type of SBCC intervention to use also depends on context, including the specific barriers to behavior change that are being addressed.

Looking forward, more research on SBCC unit costs would allow for a more nuanced analysis of the drivers of SBCC unit costs, which would be helpful to better predict unit costs, leading to more accurate budgeting and planning. Additionally, with a better understanding of

SBCC unit cost drivers, researchers could further explore the relative cost-effectiveness of SBCC approaches and better discern the best value for money with SBCC investments.

## Supporting information

**S1 Table. Variance Inflation Factors (VIF) for regressions reported in** Table 4**.** (PDF)

**S1 Dataset.** (XLS)

## Author Contributions

**Conceptualization:** Lori A. Bollinger.

**Data curation:** Lori A. Bollinger, Nicole Bellows, Rachael Linder.

**Formal analysis:** Lori A. Bollinger, Nicole Bellows.

**Investigation:** Rachael Linder.

**Methodology:** Lori A. Bollinger.

**Project administration:** Nicole Bellows.

**Validation:** Nicole Bellows, Rachael Linder.

**Writing – original draft:** Lori A. Bollinger.

**Writing – review & editing:** Lori A. Bollinger, Nicole Bellows.

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
