## [Decision Letter · Decision Letter 0]

4 Apr 2023

PONE-D-23-02671Examining the characteristics of social and behavior change communication intervention costs in low- and middle-income countries: A hedonic method approachPLOS ONE

Dear Dr. Nicole

Thank you for submitting your manuscript to PLOS ONE. After careful consideration, we feel that it has merit but does not fully meet PLOS ONE’s publication criteria as it currently stands. Therefore, we invite you to submit a revised version of the manuscript that addresses the points raised during the review process.

We look forward to receiving your revised manuscript.

Kind regards,

Guy Franck Biaou ALE, PhD

Academic Editor

PLOS ONE

“Breakthrough RESEARCH is made possible by the generous support of the American people through the United States Agency for International Development (USAID) under the terms of cooperative agreement no. AID-OAA-A-17-00018. The contents of this document are the sole responsibility of the Breakthrough RESEARCH and Population Council and do not necessarily reflect the views of USAID or the United States Government.”

“This work was funded by USAID and developed by Breakthrough RESEARCH. Breakthrough RESEARCH is made possible by the generous support of the American people through the United States Agency for International Development (USAID) under the terms of cooperative agreement no. AID-OAA-A-17-00018. The contents of this document are the sole responsibility of Breakthrough RESEARCH and do not necessarily reflect the views of USAID or the United States Government.”

“Breakthrough RESEARCH is made possible by the generous support of the American people through the United States Agency for International Development (USAID) under the terms of cooperative agreement no. AID-OAA-A-17-00018. The contents of this document are the sole responsibility of the Breakthrough RESEARCH and Population Council and do not necessarily reflect the views of USAID or the United States Government.”

4. We noted in your submission details that a portion of your manuscript may have been presented or published elsewhere. [The source data on SBCC unit costs are a subset of publicly available data at: https://breakthroughactionandresearch.org/wp-content/uploads/2021/02/BR_Cost_Repository.xlsx

Further variables were added to the data used in this analysis.] Please clarify whether this publication was peer-reviewed and formally published. If this work was previously peer-reviewed and published, in the cover letter please provide the reason that this work does not constitute dual publication and should be included in the current manuscript.

Reviewers' comments:

Reviewer's Responses to Questions

**Comments to the Author**

1. Is the manuscript technically sound, and do the data support the conclusions?

Reviewer #1: Yes

Reviewer #2: Yes

2. Has the statistical analysis been performed appropriately and rigorously? 

Reviewer #1: Yes

Reviewer #2: Yes

3. Have the authors made all data underlying the findings in their manuscript fully available?

Reviewer #1: Yes

Reviewer #2: Yes

4. Is the manuscript presented in an intelligible fashion and written in standard English?

Reviewer #1: Yes

Reviewer #2: Yes

5. Review Comments to the Author

Reviewer #1: The manuscript presents a novel hedonic pricing methodology to estimate the costs of social and behavior change communication (SBCC) interventions. This paper aims to evaluate the contribution of social and economic factors to the costs of SBCC interventions, which can help set budget expectations and has significant practical implications. However, before considering it for publication, several issues must be addressed.

Specific Comments:

1. The authors should provide more citations or explain in more detail if the hedonic pricing model has been previously used to study health-related topics.

2. A more detailed introduction of the data source is needed as the current version is insufficient and overly simplistic.

3. The authors should explain how they define "significant data quality issues" when setting the DataQuality variable.

4. Reporting the VIF results, as mentioned in the methodology, would improve the paper.

5. The authors should clarify why they chose 2005 and 2012 as cutoffs for the Technology variable.

6. The authors should include the standard deviation of each variable in Table 2.

7. Given the relatively small sample size, the authors should consider expanding the sample.

8. The authors should address the potential for over-fitting due to the high R-square value and many predictors.

9. The authors should use italics when reporting p-values in the paper.

10. Do the authors have any recommendations regarding which intervention method is better, media or IPC?

11. The authors could extend the discussion section to explain how their evaluation of SBCC costs could help in practice.

I hope these comments are helpful, and I look forward to seeing the revised manuscript.

Reviewer #2: Please check the manuscript language.

Check you the formal (all the equations).

The manuscript have a tactical sounds, but it needs to be more convenient phrases.

Check all the references and they maintain inside the manuscript.

6. PLOS authors have the option to publish the peer review history of their article (what does this mean?). If published, this will include your full peer review and any attached files.

Reviewer #1: No

Reviewer #2: **Yes: **Faris M.Alwan

---

## [Author Response · Author response to Decision Letter 0]

24 May 2023

RESPONSE TO EDITORS

1. Please ensure that your manuscript meets PLOS ONE's style requirements, including those for file naming. We have reviewed the style requirements and made the necessary adjustments. 

2. Please state what role the funders took in the study. We would like to amend our Financial Disclosure statement to add the requested underlined language:

“Breakthrough RESEARCH is made possible by the generous support of the American people through the United States Agency for International Development (USAID) under the terms of cooperative agreement no. AID-OAA-A-17-00018. The contents of this document are the sole responsibility of the Breakthrough RESEARCH and Population Council and do not necessarily reflect the views of USAID or the United States Government. The funders had no role in study design, data collection and analysis, decision to publish, or preparation of the manuscript.”

3. Please remove any funding-related text from the manuscript. We have deleted the Acknowledgements section that previously referenced the funders.

4. Please clarify whether this publication was peer-reviewed and formally published. The data underlying the analysis is available in the SBC cost repository found on the Breakthrough RESEARCH website at https://breakthroughactionandresearch.org/wp-content/uploads/2021/02/BR_Cost_Repository.xlsx. This database has not been peer-reviewed and/or formally published. 

5. Please upload your study’s minimal underlying data set as either Supporting Information files or to a stable, public repository and include the relevant URLs, DOIs, or accession numbers within your revised cover letter. We have used specific observations from the SBC cost repository and have generated additional variables to conduct this analysis. As such, we have uploaded the database as a supporting information file.

6. Please review your reference list to ensure that it is complete and correct. We have reviewed the references to ensure the list is complete and accurate. 

RESPONSE TO REVIEWERS

Many thanks to both reviewers for their thoughtful comments. We have revised the manuscript based on their suggestions; please find below detailed responses to each point raised.

Reviewer 1

1. The authors should provide more citations or explain in more detail if the hedonic pricing model has been previously used to study health-related topics.

We have added further citations on hedonic pricing models, including for health-related topics. In addition, we added more information regarding why we are using the hedonic pricing model to examine SBCC costs. Basically, while most costing studies in global health have detailed, disaggregated data available to examine cost drivers, the SBCC cost data in the literature generally do not have consistent and reliable disaggregated data available. Because our final goal was to develop an SBCC costing tool (more on this in our response to question #11 below), we decided to use a pure hedonic pricing method approach. 

2. A more detailed introduction of the data source is needed as the current version is insufficient and overly simplistic.

We have added a paragraph describing the database and citing further references. 

3. The authors should explain how they define "significant data quality issues" when setting the DataQuality variable.

Many times, the SBCC cost data are not reported thoroughly in the literature, particularly in the studies that were published earlier. When there was a lack of clarity about the reporting of various elements – especially details about the denominator – we noted in the dataset that there was a data quality issue associated with the observation, and what the issue was. In addition, sometimes we had concerns about the data quality due to issues such as small sample sizes, as well as potentially missing information (e.g., one intervention described the steps followed in delivering the intervention, but it was not clear that one of the steps was included in the final costs obtained). We have added language about this to the paper. Note that the “Data quality” dummy variable was never statistically significant in any of the regressions.

4. Reporting the VIF results, as mentioned in the methodology, would improve the paper.

Thank you for the suggestion. We have added details about the (moderate) VIF results for the sample as a whole (ranging from 1.15 to 2.76, mean VIF of 1.85) and also included a supplemental table that displays the detailed regression results (including regression-specific VIFs) for the final regressions presented in Table 4. 

5. The authors should clarify why they chose 2005 and 2012 as cutoffs for the Technology variable.

The timespan of articles included in the database ranges from 1978 to 2021. While the cutoffs for approximating periods of technology change are admittedly subjective, in examining the types of SBCC interventions implemented, we observed that the literature began to report several results for group interpersonal communication (IPC) interventions beginning in 2005, whereas there had only been one such study in the past, in 1987 - thus (arguably) a change in technology in how SBCC interventions were being delivered. Regarding the 2012 cutoff, the literature began reporting results for SMS interventions beginning in 2013, which we felt was indeed a significant change in technology. Thus, we settled on the time periods <2005, 2005-2012, and >2012. We have added some language to that effect to the paper. 

6. The authors should include the standard deviation of each variable in Table 2.

 Thank you for this suggestion; now included.

7. Given the relatively small sample size, the authors should consider expanding the sample.

We agree that a larger sample size would be preferable for conducting this analysis and note this as a limitation in the discussion section. Unfortunately, there are limited data available on SBCC unit costs, especially unit costs that use comparable denominators for analysis purposes. This dataset represents the universe of available SBCC estimates, as identified during the study period via a systematic review. As such, expanding the sample size is not feasible until more data become available through the peer reviewed or gray literature. We have added some emphasis on this in the paper, pairing it with the potential over-fitting issue noted below in #8.

8. The authors should address the potential for over-fitting due to the high R-square value and many predictors.

Thank you for your comment; we completely agree and have added this to the discussion section, pairing it with our discussion of the small sample issue (also noted above in #7).

9. The authors should use italics when reporting p-values in the paper.

Thank you for this suggestion; we have now italicized the “p” when reporting p-values.

10. Do the authors have any recommendations regarding which intervention method is better, media or IPC?

Evaluating the impact or cost-effectiveness of these interventions is beyond the scope of this paper; however, we have used these data in constructing business cases for SBCC for family planning, malaria, and nutrition (see Making the Business Case for Social and Behavior Change - Breakthrough ACTION and RESEARCH). We found that the cost-effectiveness of different types of SBCC interventions varies by context and other factors, i.e., one type is not consistently more cost-effective than another. We have added a brief sentence about this after the additional language we added discussing the SBCC costing tool (see response to #11 below). 

11. The authors could extend the discussion section to explain how their evaluation of SBCC costs could help in practice.

We completely agree! In fact, this research underpins the Excel-based SBC costing tool we have developed that is freely available online and downloadable at: https://knowledgecommons.popcouncil.org/focus_sexual-health-repro-choice/135/). We are hoping that the tool will help stakeholders (e.g., implementing partners, donors, advocates) to assist in their planning and budgeting processes. We have added language and the citation to the discussion section around this SBC costing tool and have included the caveats noted in response #10 above, that impact needs to be considered in addition to costs when planning and budgeting for SBCC.

I hope these comments are helpful, and I look forward to seeing the revised manuscript.

Reviewer #2:

1. Please check the manuscript language. 

We have reviewed the manuscript and made some small editing changes.

2. Check you the formal (all the equations).

Assuming the reviewer is asking to check the format of the equations, we have revised the equations to be clearer and more streamlined.

3. The manuscript have a tactical sounds, but it needs to be more convenient phrases.

Assuming the reviewer is asking for less technical language, we have made revisions to various sentences to make the manuscript more accessible.

4. Check all the references and they maintain inside the manuscript.

We have verified that all references have been correctly cited, including the citations added to this revision

---

## [Editor Report · Decision Letter 1]

2 Jun 2023

Examining the characteristics of social and behavior change communication intervention costs in low- and middle-income countries: A hedonic method approach

PONE-D-23-02671R1

Dear Dr.Nicole Bellows

We’re pleased to inform you that your manuscript has been judged scientifically suitable for publication and will be formally accepted for publication once it meets all outstanding technical requirements.

Kind regards,

Guy Franck Biaou ALE, PhD

Academic Editor

PLOS ONE
---

## [Editor Report · Acceptance letter]

7 Jun 2023

PONE-D-23-02671R1 

Examining the characteristics of social and behavior change communication intervention costs in low- and middle-income countries: A hedonic method approach 

Dear Dr. Bellows:

I'm pleased to inform you that your manuscript has been deemed suitable for publication in PLOS ONE. Congratulations! Your manuscript is now with our production department. 

Kind regards, 

on behalf of

Dr. Guy Franck Biaou ALE 

Academic Editor

PLOS ONE